# Development of High-Resolution Dedicated PET-Based Radiomics Machine Learning Model to Predict Axillary Lymph Node Status in Early-Stage Breast Cancer

**DOI:** 10.3390/cancers14040950

**Published:** 2022-02-14

**Authors:** Jingyi Cheng, Caiyue Ren, Guangyu Liu, Ruohong Shui, Yingjian Zhang, Junjie Li, Zhimin Shao

**Affiliations:** 1Department of Nuclear Medicine, Fudan University Shanghai Cancer Center, Department of Oncology, Shanghai Medical College, Fudan University, Shanghai 200032, China; jcheng13@fudan.edu.cn (J.C.); yingjian.zhang@sphic.org.cn (Y.Z.); 2Department of Nuclear Medicine, Shanghai Proton and Heavy Ion Center, Fudan University Cancer Hospital, Shanghai 201321, China; 3Department of Nuclear Medicine, Shanghai Proton and Heavy Ion Center, Shanghai 201321, China; rencaiyue@163.com; 4Department of Breast Surgery, Fudan University Shanghai Cancer Center, Department of Oncology, Shanghai Medical College, Fudan University, Shanghai 200032, China; liugy688@163.com; 5Key Laboratory of Breast Cancer in Shanghai, Fudan University Shanghai Cancer Center, Fudan University, Shanghai 200032, China; 6Department of Pathology, Fudan University Shanghai Cancer Center, Shanghai 200032, China; shuiruohong2014@163.com; 7Department of Oncology, Shanghai Medical College, Fudan University, Shanghai 200032, China

**Keywords:** breast cancer, axillary lymph node status, 18F-FDG dedicated PET, radiomics, machine learning

## Abstract

**Simple Summary:**

Accurate clinical axillary evaluation plays an important role in the diagnosis of and treatment planning for breast cancer (BC). This study aimed to develop a machine learning model integrating dedicated breast PET and clinical characteristics for prediction of axillary lymph node status in cT1-2N0-1M0 BC non-invasively. The performance of this integrating model in identifying pN0 and pN1 with the AUC was 0.94. We achieved an NPV of 96.88% in the cN0 and PPV of 92.73% in the cN1 subgroup. The higher true positive and true negative rate could delineate clinical subtypes and apply more precise treatment for patients with early-stage BC.

**Abstract:**

Purpose of the Report: Accurate clinical axillary evaluation plays an important role in the diagnosis and treatment planning for early-stage breast cancer (BC). This study aimed to develop a scalable, non-invasive and robust machine learning model for predicting of the pathological node status using dedicated-PET integrating the clinical characteristics in early-stage BC. Materials and Methods: A total of 420 BC patients confirmed by postoperative pathology were retrospectively analyzed. 18F-fluorodeoxyglucose (^18^F-FDG) Mammi-PET, ultrasound, physical examination, Lymph-PET, and clinical characteristics were analyzed. The least absolute shrinkage and selection operator (LASSO) regression analysis were used in developing prediction models. The characteristic curve (ROC) of the area under receiver-operator (AUC) and DeLong test were used to evaluate and compare the performance of the models. The clinical utility of the models was determined via decision curve analysis (DCA). Then, a nomogram was developed based on the model with the best predictive efficiency and clinical utility and was validated using the calibration plots. Results: A total of 290 patients were enrolled in this study. The AUC of the integrated model diagnosed performance was 0.94 (95% confidence interval (CI), 0.91–0.97) in the training set (*n* = 203) and 0.93 (95% CI, 0.88–0.99) in the validation set (*n* = 87) (both *p* < 0.05). In clinical N0 subgroup, the negative predictive value reached 96.88%, and in clinical N1 subgroup, the positive predictive value reached 92.73%. Conclusions: The use of a machine learning integrated model can greatly improve the true positive and true negative rate of identifying clinical axillary lymph node status in early-stage BC.

## 1. Introduction

The axillary lymph node (ALN) is the first station of breast lymphatic drainage [1]. Sentinel lymph node excision biopsy (SLNB) and surgical axillary lymph node dissection (ALND) are the gold standard for diagnosing pathological node status (pNx) in early-stage breast cancer (BC). However, both methods are invasive with the risk of pain, numbness, and lymphedema. In addition, a previous study showed that two-thirds of cN0 patients were diagnosed to be pN0 after SLNB [2], indicating that two-thirds of early-stage BC received overtreatment. Therefore, improving the true positive and true negative rate of clinical axillary lymph node evaluation can screen out the negative (surgical resection can be omitted), positive (ALND can be performed), and uncertain patients (SNLB can be performed); however, this may require an independent and objective tool to delineate subtypes and provide precise treatment [3].

^18^F-fluorodeoxyglucose (^18^F-FDG) positron emission tomography (PET) can provide comprehensive functional information about tumors, such as heterogeneity, metabolism, aggressiveness, and proliferation. Newer dedicated PET (D-PET), including dedicated breast PET (Mammi-PET) and dedicated axillary lymph node PET (Lymph-PET), is an advanced screening method with a spatial resolution higher than of whole-body PET/computed tomography (CT) (WB-PET/CT). Since WB-PET/CT is performed with patients in the supine position, there is a collapse of breast volume and blurring owing to respiratory motion [4,5]. In contrast, Mammi-PET comprises a single ring detector that translates axially over the length of the breast; the prone position enables full-breast volume imaging by avoiding breast compression [6]. The Lymph-PET device contains movable double-planar confronted detectors with an axilla capability view for precisely detecting hot lesions.

Compared with tissue-based biomarker testing, algorithm-based medical imaging features have inherent advantages because of being real-time, non-invasive, independent of sampling bias, and not limited to the portion of tested tissue [7]. PET radiomics features provide a complementary tool to extract high-dimensional and valuable data, such as tumor heterogeneity and shape, from images; they may be used alone or in combination with demographic, histologic, or proteomic data for clinical problem solving [7].

This study aimed to develop a scalable, non-invasive and robust machine learning model for the prediction of pNx using D-PET radiomics integrating the clinical characteristics in early-stage BC. Furthermore, we validated the potential effectiveness of this model in cN0 and cN1 subtypes to provide a positive predictive value (PPV) for cN1 and a negative predictive value (NPV) for cN0 patients.

## 2. Methods

### 2.1. Patients

This prospective study was approved by the Institutional Ethics Committee. Informed consent was obtained from each patient before participation in the study. In this study, we enrolled women (age, ≥18 years) with newly diagnosed, histologically confirmed, unilateral invasive cT1-2N0-1M0 BC. Both the primary tumor and ALN status were assessed using ultrasound (US). Tumor staging was based on the eighth edition of the American Joint Committee on Cancer staging manual [8]. The pNx was based on the microscopic assessment of at least one lymph node that was sampled using fine-needle aspiration (FNA), SLNB, or ALND, and the clinical N category (cNx) was based on physical examination (PE) and US. The cN0 was defined as no regional lymph node metastases detected on PE and US, whereas the cN1 was defined as metastases to movable ipsilateral level Ⅱ axillary lymph node(s) [8]. The patient recruitment process, exclusion criteria, and study workflow are presented in Figure 1.

Patients were randomly divided into two sets with a ratio of 7 to 3: (1) training set, on which the best-fitting prediction models were built and tested, and (2) an internal validation set, on which performance and goodness of fit were assessed. This study conformed to the transparent reporting of a multivariable prediction model for individual prognosis or diagnosis reporting guidelines [9].

### 2.2. Conventional Examination and Evaluation

All patients underwent routine PE and US, D-PET, and core needle biopsy for diagnosing invasive primary carcinoma of the breast. The PE of lymph nodes was considered positive if, on inspection and palpation, a large ALN was found, and negative if no ALN could be palpated. US findings were considered positive if ALN (of any size) was detected and negative if not. The primary lesion was evaluated using US, and its long and short diameters were recorded; the site of the primary lesion was also recorded—central, medial, lateral, and diffuse distribution.

### 2.3. Surgery Procedure and Pathological Evaluation

For determining the hormone receptor (HR), human epidermal growth factor receptor 2 (HER2) expression, and Ki-67 proliferative index, immunohistochemistry (IHC) was performed on hematoxylin and eosin-stained sections of the tissue obtained from the primary tumor using core needle biopsy. The cutoff value for estrogen receptor (ER) positivity and progestogen receptor (PR) positivity was established at 10% of the tumor cells with positive nuclear staining. The HER2 status was considered positive if on IHC, the score was 3+ or if a score of 2+ on IHC was confirmed using fluorescence in situ hybridization (FISH). HER2 copy number >6.0 or HER2/CEP17 (chromosome enumeration probe-17) ratio >2.0 was defined as FISH positive [10].

For cN0 patients, ALN was pathologically assessed using SLNB alone or SLNB and ALND. For cN1, ALN was pathologically confirmed using FNA. If FNA was negative, SLNB was performed. If FNA was positive, neoadjuvant therapy was initiated. For patients with no more than two positive lymph nodes on SLNB, the decision whether or not to perform axillary dissection depended on the operation type (breast-conserving therapy or mastectomy) and individual pathological characteristics. Based on pathological examination, ALNs were classified as macro-metastasis (>2.0 mm), micro-metastasis (0.2–2.0 mm), and isolated tumor cells (ITCs, <0.2 mm) according to the tumor-node-metastasis staging system. It should be noted that both ITCs and micro-metastases were considered negative in the final statistical analysis.

### 2.4. D-PET Examination and Evaluation

All patients were advised to fast for at least 4 h before the procedure. They were injected with 110–130 MBq ^18^F-FDG. Blood glucose levels <10 mmol/L were ensured in all patients. After a resting (tracer distribution) period of 60 min, Mammi-PET (Oncovision, Valencia, Spain) and Lymph-PET were performed sequentially.

For Mammi-PET acquisition, the patient was prone positioned on an imaging table with the breast hanging freely through an aperture in the table. The total acquisition time was 5–10 min/breast (depending on the breast length). When the bilateral breast imaging was completed, the bilateral axillary regions were scanned using Lymph-PET. For Lymph-PET, the patient sat down comfortably on a fixed chair with raised upper arm, which was supported by a dedicated bracket. The total acquisition time was 3 min/axilla.

To evaluate Lymph-PET images and quantify single-voxel maximum standard uptake value (SUVmax), commercial Medical Image Merge (version 6.5.4; MIM Software Inc., Beachwood, OH, USA), a professional image processing software certified by the United States Food and Drug Administration, was used. Two nuclear medicine physicians with 10 years of experience in PET/CT who were blinded to study-related information besides the laterality of BC analyzed the images separately. Elliptic-shaped region of interest was manually delineated, and ^18^F-FDG uptake (SUVmax) was calculated in the delineated region of interest. The highest SUVmax was selected as the study value when multiple lymph nodes were detected. The SUVmax cutoff value of lymph node on Lymph-PET was set at 0.27 according to a previous study [11]. SUVmax of ≥0.27 was considered positive, while that of <0.27 was considered negative.

### 2.5. Tumor Segmentation

Tumor was visualized and segmented on the Mammi-PET images by the aforementioned experienced nuclear medicine physicians independently using PET Edge with the MIM software. PET Edge is a gradient-based semi-automatic contouring algorithm that uses the maximum spatial gradient to detect boundaries between the tumor and normal tissue, free of different reconstruction algorithms, imaging techniques, and sphere diameter effects. It is a more accurate, consistent, and robust method for contouring tumor volumes on PET images compared with methods using visual judgment and SUV threshold [12].

### 2.6. Quantitative Radiomics Feature Extraction

Quantitative radiomics features (*n* = 851) were extracted from tumor images using the Pyradiomics package and 3D Slicer image computing platform [13,14]. All radiomics features could be extracted from both original segmented images and wavelet filtered images, except for shape features, which were independent of intensity values and therefore could only be extracted from original images. The feature extraction and its definition were in accordance with the Imaging Biomarker Standardization Initiative [15].

### 2.7. Model Development and Validation

The least absolute shrinkage and selection operator (LASSO) logistic regression with 10-fold cross-validation via minimum criteria was performed to select optimal features for predicting ALN status in the training set [16,17]. The prediction models were developed by the multivariable regression with the Akaike’s information criterion (AIC). They were then applied to differentiate pN0 from pN1 patients, and the prediction score (Pre-score) was calculated for each patient using the linear fusion of the selected non-zero features and their coefficients.

The performance of the prediction model was evaluated using the receiver operating characteristic (ROC) analysis and compared using the DeLong test in both the training and validation sets. The area under the curve (AUC) with 95% confidence interval (CI), sensitivity, specificity, accuracy, PPV, and NPV were calculated to assess model performance. The clinical utility of the models was determined and compared using the decision curve analysis (DCA) and clinical impact curve (CIC). The DCA was used for quantifying the net benefit of the patient under different threshold probabilities in the queue, and the CIC was used for estimating the number of patients who would be declared high risk for each risk threshold by the Combined Model and demonstrating the proportion of true positive patients [18].

### 2.8. Nomogram Development and Validation

We developed an individualized nomogram based on the prediction model with the highest AUC and clinical utility to provide a visually quantitative tool for predicting the ALN status in early-stage BC patients in the training set [19]. Calibration curves, reflecting the agreement between the predicted probability of the nomogram and actual probability, were plotted using 1000 bootstrap resamples based on the internal (training set) and external (validation set) validity.

### 2.9. Statistical Analysis

Univariate and multivariate analyses were performed using R software (version 4.0, http://www.r-project.org accessed on 15 June 2021). The comparison between the two groups was performed using Fisher’s exact test or χ^2^ test for categorical variables and independent *t*-test or Mann-Whitney *U* test for continuous variables. A two-sided *p* < 0.05 indicated statistical significance. Intra- and inter-class correlation coefficients (ICCs) were used to evaluate the consistency and reproducibility of the intra- and inter-observer agreement of the radiomics features. An ICC value of >0.75 indicated good reliability.

## 3. Results

### 3.1. Demographic and Clinicopathological Characteristics

We screened 420 patients with cT1-2N0-1M0 BC between 9 September 2019 and 30 September 2020. Finally, a total of 290 women (mean age, 50.46 ± 10.58 years; range, 28–78 years) with invasive lesions (invasive ductal carcinoma, 283; invasive lobular carcinoma, 7) were enrolled in this study. The patients’ demographic and clinicopathologic characteristics were separately compared between the training (*n* = 203, 70.00%) and validation (*n* = 87, 30.00%) sets to identify potential diagnostic biomarkers for the ALN status (Table 1).

There were no significant differences in primary tumor’s site, location, ER level, and HER-2 status between the pN0 and pN1 groups in the univariate analysis (*p* > 0.05). The findings on Lymph-PET, US, and PE were significantly related to the ALN status in both the training and validation sets (*p* < 0.05).

### 3.2. Feature Selection and Model Development

A total of 851 radiomics features comprising shape (*n* = 14), first-order statistics (*n* = 18), texture (*n* = 75; 24 gray level co-occurrence matrix (GLCM) features, 14 gray level dependence matrix (GLDM) features, 16 gray level run length matrix (GLRLM) features, 16 gray level size zone matrix (GLSZM) features, and 5 neighboring gray tone difference matrix (NGTDM) features), and wavelet features obtained from the filters (H: High pass filter, L: Low pass filter) applied in the x, y, z directions (*n* = 744) were separately extracted from the tumor regions with increased ^18^F-FDG uptake segmented by two nuclear medicine physicians, and minutely described in Appendix A.

Initially, 7 of the 12 clinicopathologic markers, 34 of the 851 radiomics features, and 19 of the 864 combined features (Lymph-PET finding, 1; clinicopathologic markers, 12; and radiomics features, 851) were separately selected by the LASSO regression (Figure 2). Subsequently, three independent prediction models were developed using the most valuable 4 clinicopathologic markers, 10 Mammi-PET radiomics parameters, and 11 combined features selected by the multivariable regression with the AIC for differentiating pN0 from pN1 patients in the training set. For all prediction models, pN1 BC patients generally had higher pre-scores calculated using the following formulas than pN0 BC patients (*p* < 0.05) (Figure 3, Table 2).

Pre-score (Clinicopathologic Model) = −4.30 + 1.93 × US (Negative: 0, Positive: 1) + 2.01 × PE (Negative: 0, Positive: 1) + 0.74 × Tumor Location (Central: 0, Medial: 2, Lateral: 3, Diffuse: 4) + 0.02 × Ki-67.

Pre-score (Radiomics Model) = −0.38 + 4.27 × 10^−3^ × Wavelet-LLH (WLLH)_Median + 33.06 × WLLH_GLDM_ Dependence Variance + 0.04 × WLHH_Mean −0.01 × WHLL_Median + 106.7 × WHLL_NGTDM_Busyness −601.16 × WHLH_GLDM_Large Dependence Low Gray Level Emphasis −0.05 × WHHL_Mean + 0.05 × WHHL_Median + 5.31 × WHHH_GLCM_MCC −123.61 × WLLL_GLDM_Dependence Variance.

Pre-score (Combined Model) = −8.23 +1.78 × Lymph-PET (Negative: 0, Positive: 1) + 2.44 × US (Negative: 0, Positive: 1) + 1.92 × PE (Negative: 0, Positive: 1) + 1.08 × Tumor Location (Central: 0, Medial: 2, Lateral: 3, Diffuse: 4) + 0.02 × ER + 0.03 × Ki67 −2.55 × WLLH_Skewness + 42.51 × WLHH_GLCM_Correlation + 0.03 × WHHL_Median −6.44 × 10^−9^ × WHHH_GLCM_Cluster Prominence −0.84 × WHHH_GLSZM_Zone Variance.

### 3.3. Model Performance and Clinical Utility

The Clinicopathological Model contained US and PE achieved the higher AUC values of 0.88 and 0.84 in the training and validation sets, respectively, while separate US and PE achieved AUC values of 0.77 and 0.77 in the training set and 0.77 and 0.70 in the validation set, respectively (*p* < 0.05). The Radiomics Model comprised of 10 Mammi-PET radiomic parameters performed better in the training set (AUC: 0.81) than in the validation set (AUC: 0.65).

According to the DeLong test, the Combined Model, which comprised six clinicopathologic factors and five Mammi-PET radiomics parameters, showed the highest AUC, best predictive accuracy, and NPV among the three models in both the training set (AUC: 0.94, accuracy: 87.68%, NPV: 84.85%, *p* < 0.05) and validation set (AUC: 0.93, accuracy: 87.36, NPV: 93.18%, *p* < 0.05). The detailed statistical results of the models’ performance in discriminating pN0 from pN1 patients are summarized in Table 3, and its corresponding ROCs are shown in Figure 4a,b.

The DCA showed that the Combined Model was the most reliable and valuable tool to predict the ALN status when the ALN metastasis (ALNM) threshold probability was greater than 10% (Figure 4c). The CIC of the Combined Model presented the risk stratification in predicting 1000 people, including the estimated number of people who would be declared a high risk for ALN metastasis and true positive cases under each threshold probability (Figure 4d).

Furthermore, we validated the potential effectiveness of the Combined Model in the total dataset. Among 290 patients, 107 (36.90%) had cN0 and the remaining 183 (63.10%) had cN1 on the basis of baseline clinical and imaging data. The Combined Model was highly effective in identifying pN0 from cN0 patients with an AUC of 0.83 and an NPV of 96.88%, while identifying pN1 from cN1 patients with an AUC of 0.90 and a PPV of 92.73% (Figure 4e,f, Table 4).

### 3.4. Nomogram Development and Validation

With the results above, we developed an individualized nomogram using the Combined Model’s risk features for visualization (Figure 5a). Then, the risk probability of ALNM for each patient could be calculated directly according to the nomogram. The optimal threshold to discriminate between pN0 and pN1 was 0.59. The calibration curves demonstrated a good agreement between the prediction probability of ALNM by the nomogram and the actual observation of ALN metastasis in both the training and validation sets (Figure 5b,c).

## 4. Discussion

Routine primary lesion and ALN status assessment includes US, mammography (MMG), and magnetic resonance imaging (MRI). Although US remains one of the key tools for ALN assessing, it has the limitation of being subjective [20]. A systematic review analyzing the use of US showed significant variation among institutions, with overall sensitivity and specificity ranging between 26% and 76% and between 88% and 98%, respectively [21]. A more recent meta-analysis involving 21 studies found that the assessment of abnormal nodes using US had a median sensitivity and specificity of 64% and 82%, respectively [22]. Although MMG is suitable for examining breast tissue, it is not considered reliable for ALN evaluation because a part of the axillary area might not be visible on routine MMG [23]. MRI is used to assess newly diagnosed BC and examine the response to neoadjuvant treatment; however, it may also provide insufficient imaging of the axillary region [24]. Thus, the routine non-invasive techniques for assessing ALN status have a much lower accuracy for N0-1 identification in early-stage BC.

Recently, the WB-PET-based radiomics model showed promising results in predicting occult lymph node metastasis in cN0 tumors, including lung cancer, cervical cancer, and esophageal adenocarcinoma [25,26,27]. Hasan et al. analyzed the textural features of WB-PET/CT coming from 124 breast cancer patients and showed the gray-level zone length matrix (GLZLM) could be the predictive parameter of ALN, but the AUC was only 0.64 [28]. Similarly, in the study by Bong [29], the primary tumor of 100 invasive ductal BC patients was analyzed using the WB-PET/CT-based radiomics model to predict ALN status. The results showed that the AUC, sensitivity, specificity, and accuracy of the Radiomics Model in predicting ALN metastasis were 0.890, 90.9%, 71.4%, and 80%, respectively. These studies indicate that ^18^F-FDG PET-based radiomics model-derived biomarkers can enable lymph node assessment that is non-invasive, repeatable, and independent of sampling bias. Furthermore, the newer dedicated breast PET has demonstrated higher spatial resolution and uptake sensitivity in lesions [30]. First, limited spatial resolution and partial volume effects constitute a challenge for small lesions. Owing to the lower post-reconstruction voxel resolution (4 × 4 × 4 mm) and the smaller breast tumor tissue size relative to the total field of view, the primary tumor in WB-PET comprises a relatively small fraction of the total voxel volume. Hence, Hatt et al. limited their analysis to metabolically active volumes >3 cm^3^ [31]. In contrast, D-PET had a higher in-tumor resolution (1 × 1 × 1 mm post-reconstruction) that greatly decreased the threshold for tumor volume to 0.064 cm^3^ [32]. Consequently, D-PET showed an accuracy comparable to MRI and improved sensitivity comparable to WB-PET for quantifying primary lesions [33]. Second, the overall ^18^F-FDG uptake values (SUVmax, SUVmean, and SUVpeak) in the lesion were higher with D-PET than with WB-PET, indicating that with D-PET an image with a higher signal-noise ratio could be obtained and relatively lower active lesions could be detected. Finally, the improvement in spatial resolution highlighted the spatial heterogeneity within the primary breast tumor. The observed qualitative differences in spatial and signal intensity heterogeneity in D-PET may be largely driven by the higher voxel resolution and tumor tissue fraction. In a comparative study, spatial heterogeneity features showed statistically significant differences between D-PET and WB-PET [6]. The precise quantification of tumor heterogeneity may allow an accurate prediction of ALN metastasis. In summary, high-resolution D-PET could detect smaller and lower active lesions, thereby making radiomics analysis more feasible and reliable.

In earlier studies, the AUC for predicting ALNM ranged from 0.90 to 0.92 when using MRI-based radiomics [34,35] and from 0.89 to 0.90 when using US [36,37]. A retrospective study analyzed US features of 1328 cT1-2N0 BC and established nomograms for ALNM prediction. The AUC of the prediction model and external validation group was 0.802 and 0.73, respectively [38]. Another study using deep learning algorithms based on US images established an ALNM prediction model and got AUC of 0.805 [39]. Other studies also attempted to establish more excellent prediction non-invasively using clinical feature, pathological type, molecular subtype, and radiological data. However, the AUCs were only about 0.74 to 0.83 [40,41,42]. Notably, this integrated model enabled an objective and unbiased assessment; it may help in the clinical stratification of lesions for better treatment planning. In the cN0 subgroup (*n* = 107), where avoiding invasive assessment of ALN could be beneficial, the NPV was 96.88%, thereby indicating that this algorithm may have the potential to screen out patients in whom axillary surgery can be avoided. To our knowledge, no non-invasive method could achieve such a high NPV. Meanwhile, in the cN1 subgroup (*n* = 183), where ALNs were assessed using US and PE, the biggest concern was to identify the true positive node. Using the integrated model, we achieved a high PPV of 92.73%. These encouraging results show that a machine learning integrated model based on radiomics could independently predict lymph node status, modify clinical decisions, or affect patient outcomes ‘over and above’ conventional approaches.

This study had some limitations. First, this study was conducted at a single center and had a retrospective design, which could have led to a selection bias. Second, we excluded patients with multifocal breast lesions and bilateral disease because it was difficult to determine the lesion that would lead to ALN metastasis. Third, although internal validation was performed in the test cohort, validation in an external cohort was required to evaluate the transferability of the radiomics model. In addition, further controlled prospective studies are necessary to refine the predictive accuracy of this integrated model.

## 5. Conclusions

In this study, we developed a machine learning integrated model based on radiomics of ^18^F-FDG Mammi-PET, US, PE, Lymph-PET, and clinical characteristics for non-invasively identifying pNx of ALN in early-stage BC (cT1-2N0-1M0). The AUC was 0.94 (95% CI, 0.91–0.97). Using our integrated model, we achieved an NPV of 96.88% in the cN0 subgroup and a PPV of 92.73% in the cN1 subgroup. The use of the machine learning integrated model can greatly improve the true positive and true negative rate of identifying ALNM to delineate clinical subtypes and deliver precise treatment to patients with early-stage BC.

## Figures and Tables

**Figure 1 cancers-14-00950-f001:**
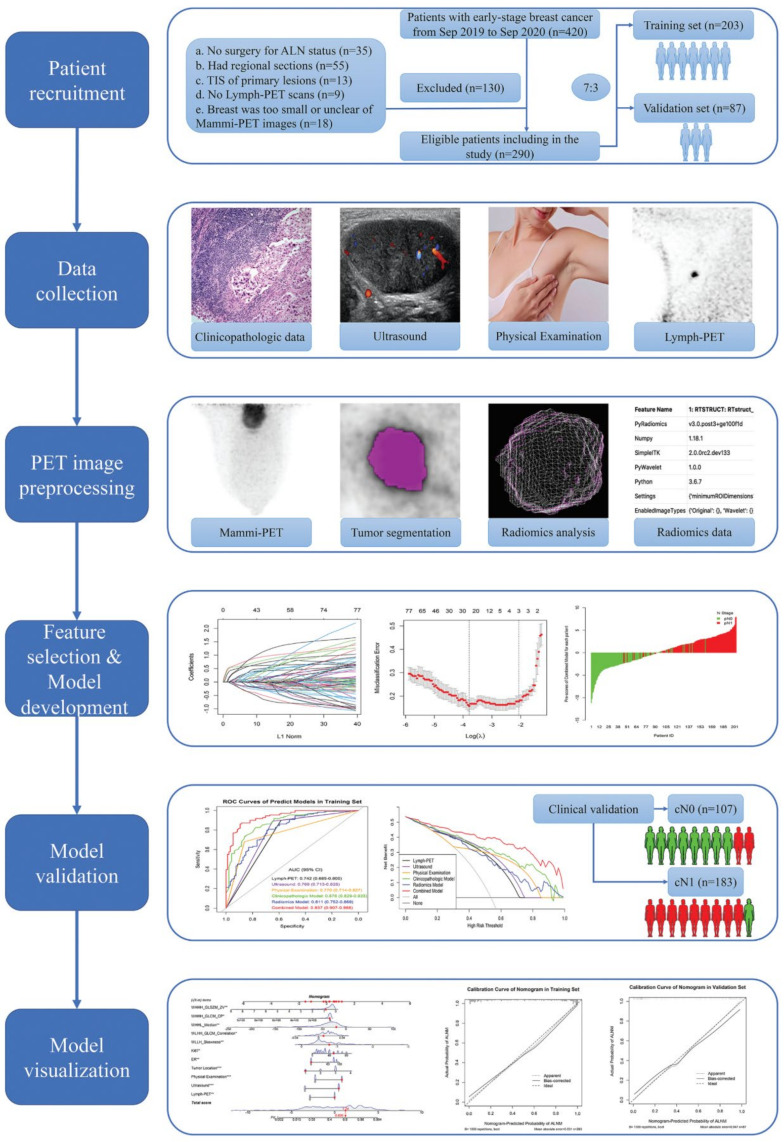
Patient recruitment process and study workflow.

**Figure 2 cancers-14-00950-f002:**
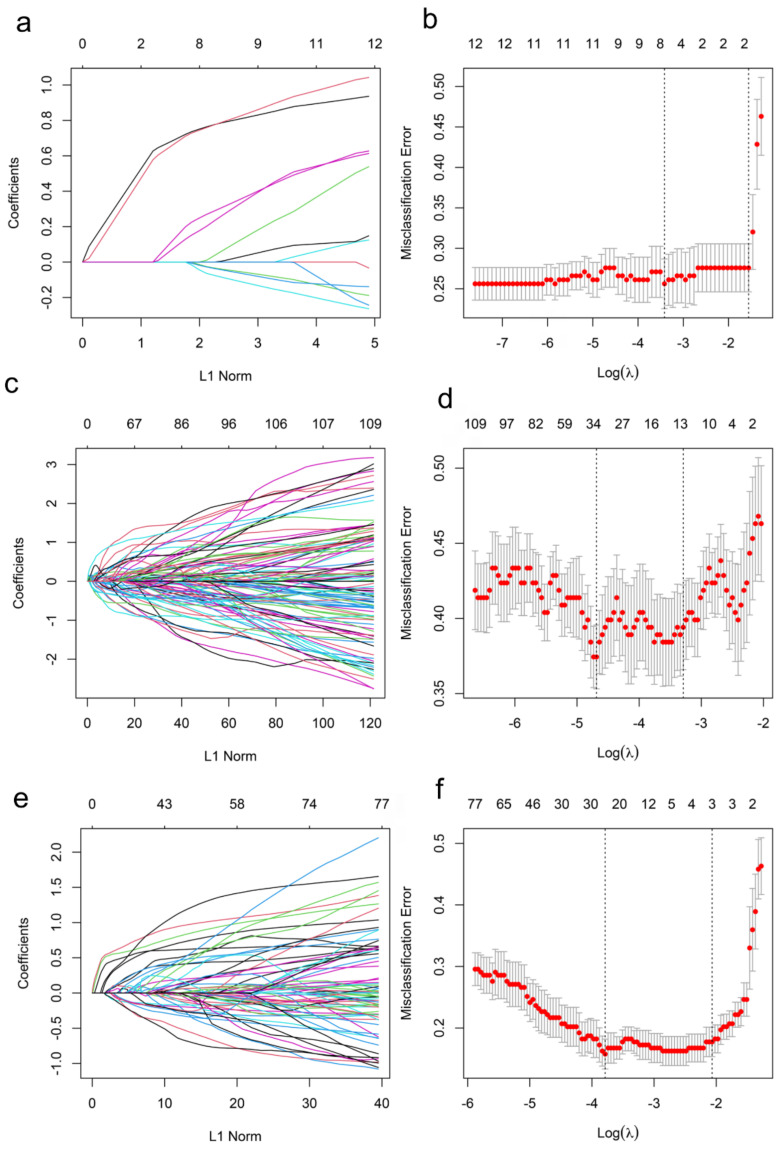
Features selection for the prediction of the models in the training set. (**a**,**c**,**e**) show the LASSO coefficient profiles of features. (**b**,**d**,**f**) show the feature selection by the LASSO model with tuning parameter (λ) using 10-fold cross-validation via minimum criteria. The X-axis shows log (λ), and the Y-axis shows the model misclassification rate. The dotted vertical lines are drawn at the optimal values using the minimum criteria and the 1-se criteria, respectively. The 7, 34, and 19 features with non-zero coefficients are initially indicated with the optimal λ values of 0.03, 0.01, 0.02 for Clinicopathologic Model (**b**), Radiomics Model (**d**), and Combined Model (**f**), respectively.

**Figure 3 cancers-14-00950-f003:**
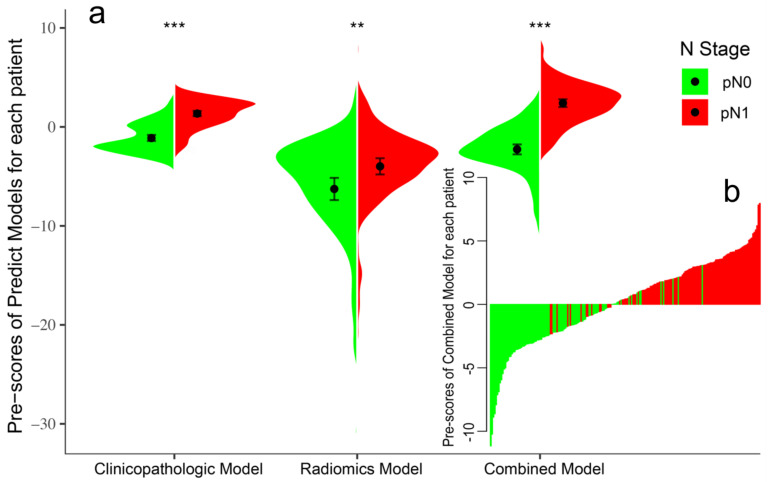
Violin plot of predict models for pN0 and pN1 patients in the training set (**a**). The black dot represents the median. The black line is the range from the lower quartile to the upper quartile. The waterfall plot of the Combined Model was used to visualize the distribution of the pre-scores of individual pN0 and pN1 patients (**b**). The ** represents. *p* value < 0.01, *** represents. *p* value < 0.001.

**Figure 4 cancers-14-00950-f004:**
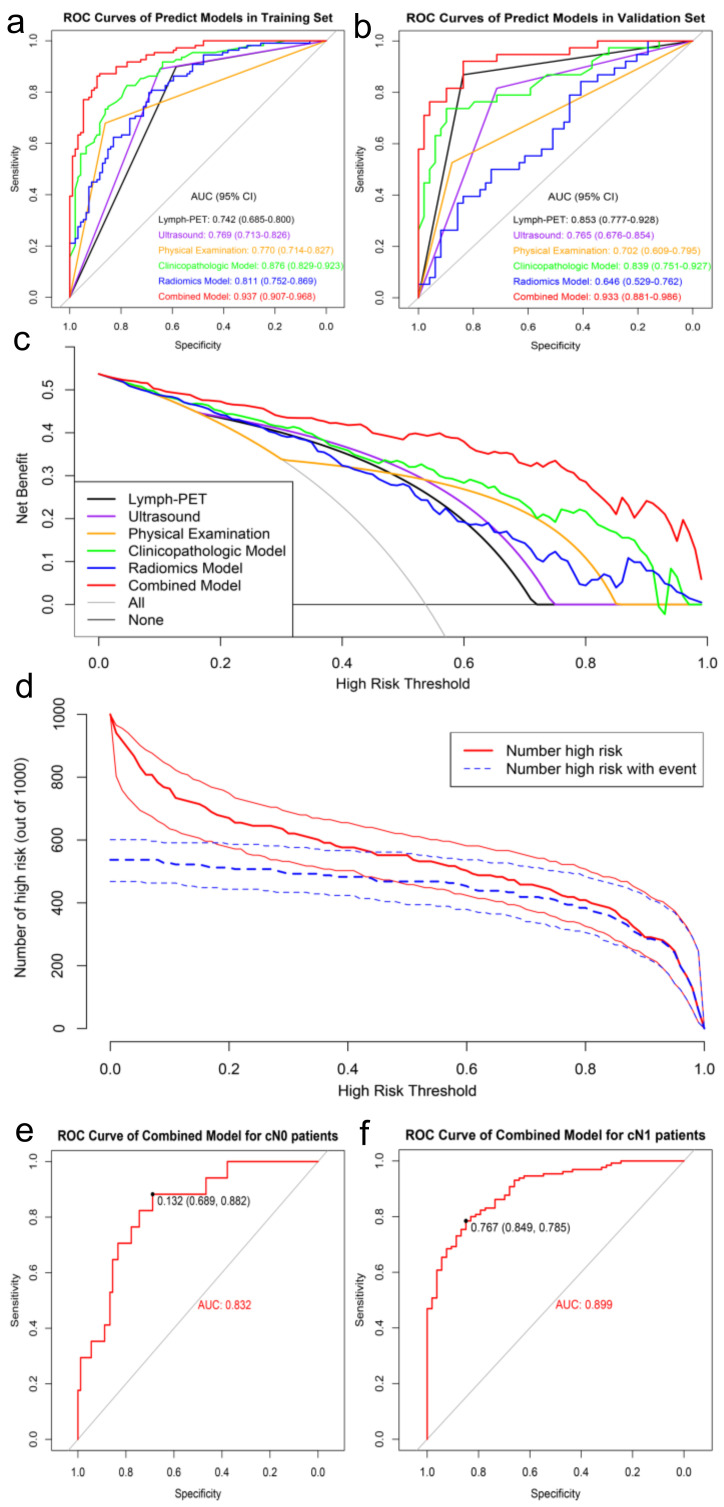
ROC analysis of predict models for predicting ALN status in the training set (**a**) and validation set (**b**), respectively. DCA of predict models in training set (**c**). The X-axis represents the threshold probability that the expected benefit of treatment was equal to the expected benefit of avoiding treatment. The Y-axis represents the net benefit. The gray and black line represent the hypothesis that all early-stage breast cancer patients were pN1 and pN0, respectively. CIC showed the Combined Model’s estimated number that would be declared high risk for each risk threshold and the proportion of true positive patients (**d**). ROC analysis of Combined Model in the total data (**e**,**f**).

**Figure 5 cancers-14-00950-f005:**
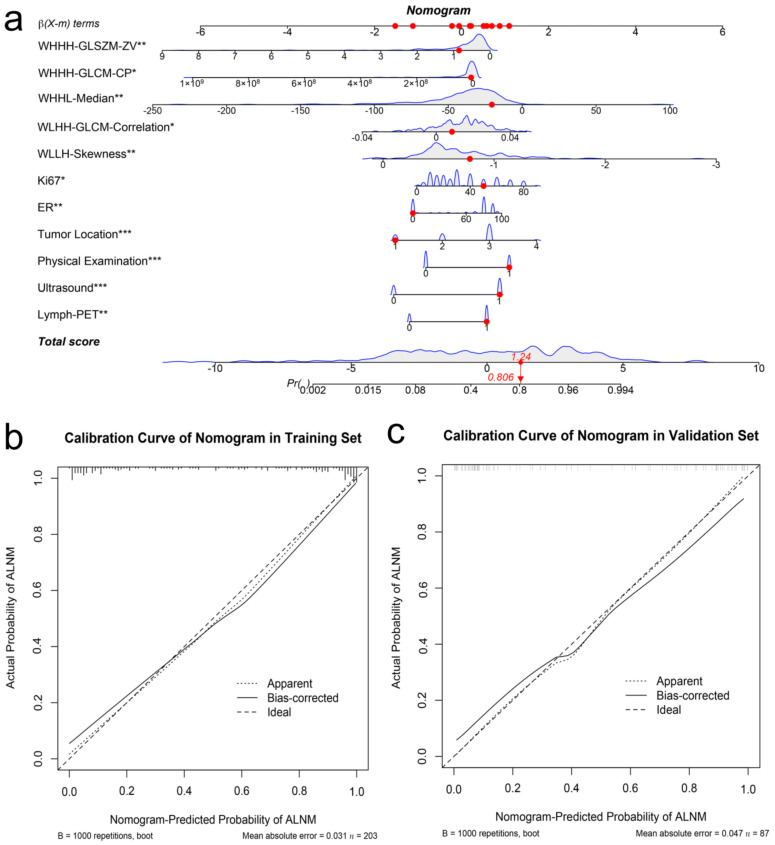
Developed the predict nomogram based on the Combined Model in the training set (**a**). The nomogram presented the ALNM probability of an early-stage breast cancer patients was 0.806, which was confirmed as pN1 stage by surgery. Calibration curves of nomogram in the training set (**b**) and validation set (**c**), respectively. The * represents. *p* value < 0.05, the ** represents. *p* value < 0.01, *** represents. *p* value < 0.001. The X-axis represents the predicted probability of ALNM estimated by nomogram, whereas the Y-axis represents the actual ALNM rates. The solid line represents the ideal reference line that predicted ALN status corresponds to the actual outcome, the short-dashed line represents the apparent prediction of nomogram, and the long-dashed line represents the ideal estimation. Calibration curves show that the actual probability corresponded closely to the prediction of nomogram.

**Table 1 cancers-14-00950-t001:** Demographic and clinicopathologic characteristics of early-stage breast cancer patients.

Characteristics	Training Set (*n* = 203)		*p*	Validation Set (*n* = 87)		*p*
	pN0 (*n* = 94)	pN1 (*n* = 109)		pN0 (*n* = 49)	pN1 (*n* = 38)	
Age (y)	53.04 ± 9.02 ^#^	49.29 ± 11.21 ^#^	0.01	49.31 ± 10.56 ^#^	48.89 ± 11.55 ^#^	0.86
Weight (Kg)	60.07 ± 9.28 ^#^	57.21 ± 7.40 ^#^	0.02	61.64 ± 7.01 ^#^	59.29 ± 8.59 ^#^	0.16
Tumor Side			0.30			0.37
Left breast	44 (46.81)	59 (54.13)		21 (42.86)	20 (52.63)	
Right breast	50 (53.19)	50 (45.87)		28 (57.14)	18 (47.37)	
Tumor Location			0.11			0.80
Central	21 (22.34)	18 (16.52)		3 (6.12)	4 (10.53)	
Medial	24 (25.54)	22 (20.18)		17 (34.70)	8 (21.05)	
Lateral	48 (51.06)	66 (60.55)		26 (53.06)	24 (63.16)	
Diffuse	1 (1.06)	3 (2.75)		3 (6.12)	2 (5.26)	
LD (mm)	23.40 ± 11.05 ^#^	32.92 ± 17.09 ^#^	<0.01	20.81 ± 10.60 ^#^	29.41 ± 14.32 ^#^	<0.01
SD (mm)	14.52 ± 6.20 ^#^	18.47 ± 8.87 ^#^	<0.01	15.38 ± 7.21 ^#^	22.88 ± 12.16 ^#^	<0.01
ER	80.00 (0.00, 90.00) *	80.00 (0.00, 80.00) *	0.23	80.00 (0.00, 80.00) *	80.00 (0.00, 90.00) *	0.72
PR	60.00 (0.00, 80.00) *	10.00 (0.00, 80.00) *	**0.01**	60.00 (0.00, 80.00) *	22.50 (0.00, 80.00) *	0.51
HER-2			0.87			0.01
Negative	64 (68.09)	73 (66.97)		41 (83.67)	22 (57.89)	
Positive	30 (31.91)	36 (33.03)		8 (16.33)	16 (42.11)	
Ki-67	20.00 (10.00, 40.00) *	30.00 (25.00, 50.00) *	<0.01	30.00 (12.50, 62.50) *	30.00 (20.00, 50.00) *	0.66
Lymph-PET			<0.01			<0.01
Negative	55 (58.51)	11 (10.09)		41 (83.67)	5 (13.16)	
Positive	39 (41.49)	98 (89.91)		8 (16.33)	33 (86.84)	
US			<0.01			<0.01
Negative	61 (64.89)	12 (11.01)		35 (71.43)	7 (18.42)	
Positive	33 (35.11)	97 (88.99)		14 (28.57)	31 (81.58)	
PE			<0.01			<0.01
Negative	81 (86.17)	35 (32.11)		43 (87.76)	18 (47.37)	
Positive	13 (13.83)	74 (67.89)		6 (12.24)	20 (52.63)	

Note: Data in parentheses were percentages unless otherwise noted. ^#^ Values refer to mean ± standard deviation. * Values refer to median (interquartile range). *p* values were derived from the univariate analysis between each of characteristic and ALN status, and the bold ones indicate statistical significance. LD: long diameter of tumor; SD: short diameter of tumor.

**Table 2 cancers-14-00950-t002:** Pre-scores of predict models for early-stage breast cancer patients in training set.

	pN0 (*n* = 94)	pN1 (*n* = 109)	*p*
Clinicopathologic Model	−1.66 (−2.28, 0.07)	1.68 (0.41, 2.42)	<0.01
Radiomics Model	−4.54 (−8.48, −2.72)	−3.39 (−5.37, −1.59)	<0.01
Combined Model	−2.19 (−3.42, −0.68)	2.64 (1.13, 3.77)	<0.01

Note: Values refer to median (interquartile range).

**Table 3 cancers-14-00950-t003:** Performance of predict models for predicting ALN status in early-stage breast cancer patients.

Training Set	AUC (95% CI)	SEN (%)	SPE (%)	ACC (%)	PPV (%)	NPV (%)
Lymph-PET	0.74 (0.68–0.80)	89.91	58.51	75.37	71.53	83.33
US	0.77 (0.71–0.83)	88.99	64.89	77.83	74.62	83.56
PE	0.77 (0.71–0.83)	67.89	86.17	76.35	85.06	69.83
Clinicopathologic Model	0.88 (0.83–0.92)	82.57	77.66	80.30	81.08	79.35
Radiomics Model	0.81 (0.75–0.87)	79.82	69.14	74.88	75.00	74.71
Combined Model	0.94 (0.91–0.97)	86.24	89.36	87.68	90.38	84.85
**Validation Set**	**AUC (95% CI)**	**SEN (%)**	**SPE (%)**	**ACC (%)**	**PPV (%)**	**NPV (%)**
Lymph-PET	0.85 (0.78–0.93)	86.84	83.67	85.06	80.49	89.13
US	0.77 (0.68–0.85)	81.58	71.43	75.86	68.89	83.33
PE	0.70 (0.61–0.79)	52.63	87.76	72.41	76.92	70.49
Clinicopathologic Model	0.84 (0.75–0.93)	73.68	89.80	82.76	84.85	81.48
Radiomics Model	0.65 (0.53–0.76)	84.21	40.82	59.78	52.46	76.92
Combined Model	0.93 (0.88–0.99)	92.11	83.67	87.36	81.40	93.18

Note: AUC: area under the receiver operating curve; CI: confidence interval; SEN: sensitivity; SPE: specificity; ACC: accuracy; PPV: positive predictive value; NPV: negative predictive value; US: Ultrasound; PE: Physical Examination.

**Table 4 cancers-14-00950-t004:** Clinical validation of Combined Model for predicting ALN status in total data.

	cN0 (*n* = 107)		cN1 (*n* = 183)	
	pN0 (*n* = 90)	pN1 (*n* = 17)	pN0 (*n* = 53)	pN1 (*n* = 130)
AUC (95% CI)	0.83 (0.73–0.93)		0.90 (0.85–0.94)	
SEN (%)	88.24		78.46	
SPE (%)	68.89		84.91	
ACC (%)	71.96		80.33	
PPV (%)	34.88		92.73	
NPV (%)	96.88		61.64	

Note: AUC: area under the receiver operating curve; CI: confidence interval; SEN: sensitivity; SPE: specificity; ACC: accuracy; PPV: positive predictive value; NPV: negative predictive value.

## Data Availability

The datasets used and/or analyzed during the current study are available from the corresponding author on reasonable request.

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
