# Peer review of "Development of High-Resolution Dedicated PET-Based Radiomics Machine Learning Model to Predict Axillary Lymph Node Status in Early-Stage Breast Cancer"

_cancers, 2022, doi:10.3390/cancers14040950_

Round 1

Reviewer 1 Report

Manuscript ID: Cancers - 1528297

Title: Development and Validation of High-resolution Dedicated Positron Emission Tomography-based Radiomics Machine Learning Model to Predict Axillary Lymph Node Status in Early-stage Breast Cancer

Referee’s comments

  1. General comment

          The present paper attempts to demonstrate the benefits of PΕΤ radiomics model for the prediction of ALD status in early stage breast cancer. The manuscript is well written, the topic is interesting providing a plethora of data and information. I suggest the publication of the manuscript, however, before publication, I propose a few minor but crucial modifications that enhance the readership of the manuscript. First of all, I suggest to shorten the title because it is too long in its present form. In addition, sometimes the reader is lost within the huge amount of information, so another important issue is the re-structure of the manuscript. For example try to: 1) develop subsections in the Methods section dividing to: a) Tumor Examination, b) Tumor segmentation and surgery procedure and c) Statistical analysis and model development and validation, or modify the aforementioned subtitles properly, 2) Information regarding the patients are provided in both Methods and Results sections. You can summarize this information in one single table or figure. The information is lost within the text. Some of the information is provided in Figure 1, but you may reduce that part in the figure and the corresponding text and can create a new one with further data (including eligible and excluded criteria, the total number of patients, age criteria (>18) the period, the divided teams etc.), 3) Create a list of abbreviations used in the manuscript. These are several terms, parameters and factors used in the manuscript which confuse the reader, 4) Transfer and unify text in lines 375-398 of the Discussion section to the Conclusions section.  

Below a few specific comments are recommended.

  1. Specific comments

As my mother tongue is not English i do not propose major language modifications but i feel that some editing would be useful. Please read again the manuscript carefully and make corrections where is necessary.  A few modifications are addressed below:

  1. Abstract: Line 23: Change: “…model for predicting of pathological node status…” to “…model for predicting the pathological node status…”.
  2. Abstract: Line 28: Change: “…The area under receiver-operator characteristic (ROC) curve (AUC)…” to “…The characteristic curve (ROC) of the area under receiver-operator (AUC)…”.
  3. Introduction: Line 70: Change: “…learning model for prediction of…” to “…learning model for the prediction of…”.
  4. Introduction: Lines 70-71 need rephrasing
  5. Methods: Line 129: Change: “…while that <0.27…” to “…while that of<0.27…”.
  6. Methods: Lines 140, 165, 177: Remove one dot from the following: “platform (12,13)..”, “set (15,16)..”. and “patients (17)..”.
  7. Methods: Line 169: Change: “…The performance of prediction models were…” to “…The performance of the prediction models were…”.
  8. Put a space between line 213 and line 214.
  9. Figure 2. Line 241: Change: “Features selection for predict models…” to “Features selection for the prediction of the models…”.
  10. Line 250. Separate the two words: “ pN1patients”.
  11. Put a space between line 252 and line 253.
  12. Figure 4. Line 282 needs rephrasing
  13. Figure 4. Line 286: Change: “…number who would be declared…” to “…number that would be declared…”
  14. Line 286: Change: “…number who would be declared…” to “…number that would be declared…”
  15. Put a space between line 291 and line 292.
  16. Put a space between line 306 and line 307.
  17. Line 323: Change: “…in the diagnosis of and treatment planning…” to “…in the diagnosis of the treatment planning…”.
  18. Line 379: Please remove or close the open parenthesis: “…(95%...”.

A few additional changes are listed below:

1) Introduction: Add a reference in Line 55 regarding the higher spatial resolution. Maybe reference 29 as cited in the Discussion section.

2) Change the y-axis of figures 2 from “Coefficients” to “LASSO coefficients”.

3) Figure 4. In subfigure (d) change y-axis from “Number high risk” to “Number of high risk”.

Author Response

  1. General comment

The present paper attempts to demonstrate the benefits of PΕΤ radiomics model for the prediction of ALD status in early stage breast cancer. The manuscript is well written, the topic is interesting providing a plethora of data and information. I suggest the publication of the manuscript, however, before publication, I propose a few minor but crucial modifications that enhance the readership of the manuscript. First of all, I suggest to shorten the title because it is too long in its present form. In addition, sometimes the reader is lost within the huge amount of information, so another important issue is the re-structure of the manuscript. For example try to: 1) develop subsections in the Methods section dividing to: a) Tumor Examination, b) Tumor segmentation and surgery procedure and c) Statistical analysis and model development and validation, or modify the aforementioned subtitles properly, 2) Information regarding the patients are provided in both Methods and Results sections. You can summarize this information in one single table or figure. The information is lost within the text. Some of the information is provided in Figure 1, but you may reduce that part in the figure and the corresponding text and can create a new one with further data (including eligible and excluded criteria, the total number of patients, age criteria (>18) the period, the divided teams etc.), 3) Create a list of abbreviations used in the manuscript. These are several terms, parameters and factors used in the manuscript which confuse the reader, 4) Transfer and unify text in lines 375-398 of the Discussion section to the Conclusions section. 

Response: Thank you for your high comment on our work. We have made the following modifications in the revised manuscript according to your suggestion:

  1. We have revised the title according to your suggestion. The title now reads as follows: “Development of High-resolution Dedicated PET-based Radiomics Machine Learning Model to Predict Axillary Lymph Node Status in Early-stage Breast Cancer” (Page 1, lines 2 to 4).
  2. We have also re-structured the Methods section of the revised manuscript. The subsections now read as follows: “a) Patients; b) Conventional examination and evaluation; c) Surgery procedure and pathological evaluation; d) D-PET examination and evaluation; e) Tumor segmentation; f) Quantitative radiomics feature extraction; g) Model development and validation; h) Nomogram development and validation; i) Statistical analysis”
  3. We have reduced the repetitive text of patient information in the Methods and Results sections.
  4. We have added a list of abbreviations after the Conclusions section (Page 21-22, lines 485-503), and made necessary adjustments to the order of abbreviations and their full name of the revised manuscript.
  5. We have transferred and unified text in lines 375-398 of the Discussion section to the Conclusions section of the revised manuscript. (Page 20-21, lines 440-483).

Below a few specific comments are recommended.

  1. Specific comments

As my mother tongue is not English i do not propose major language modifications but i feel that some editing would be useful. Please read again the manuscript carefully and make corrections where is necessary. A few modifications are addressed below:

  1. Abstract: Line 23: Change: “…model for predicting of pathological node status…” to “…model for predicting the pathological node status…”.

Response: Thank you for your suggestion. We have added the word “the” in the Abstract section of the revised manuscript (Page1, lines 32).

  1. Abstract: Line 28: Change: “…The area under receiver-operator characteristic (ROC) curve (AUC)…” to “…The characteristic curve (ROC) of the area under receiver-operator (AUC)…”.

Response: We have revised the sentence according to your suggestion (Page1, lines 37-38).

  1. Introduction: Line 70: Change: “…learning model for prediction of…” to “…learning model for the prediction of…”.

Response: We have added the word “the” in the revised manuscript (Page2, lines 86).

  1. Introduction: Lines 70-71 need rephrasing.

Response: We have rephrased the sentence in the revised manuscript as follows: “This study aimed to develop a scalable, non-invasive and robust machine learning model for the prediction of pNx, which was termed radiomics lymph node status (rNx) using D-PET radiomics integrated the clinical characteristics in early-stage BC. ” (Page2, lines 85-87).

  1. Methods: Line 129: Change: “…while that <0.27…” to “…while that of<0.27…”.

Response: We have added the word “of” in the revised manuscript (Page7, lines 172).

  1. Methods: Lines 140, 165, 177: Remove one dot from the following: “platform (12,13)..”, “set (15,16)..”. and “patients (17)..”.

Response: We have removed those redundant dots.

  1. Methods: Line 169: Change: “…The performance of prediction models were…” to “…The performance of the prediction models were…”.

Response: We have added the word “the” in the revised manuscript (Page 8, lines 214).

  1. Put a space between line 213 and line 214.

Response: We have added a space in the corresponding position.

  1. Figure 2. Line 241: Change: “Features selection for predict models…” to “Features selection for the prediction of the models…”.

Response: We have revised the sentence according to your suggestion (Page 13, lines 298).

  1. Line 250. Separate the two words: “ pN1patients”.

Response: We have added a space in the corresponding position.

  1. Put a space between line 252 and line 253.

Response: We have added a space in the corresponding position.

  1. Figure 4. Line 282 needs rephrasing.

Response: We have rephrased the sentence in the revised manuscript as follows: “The X-axis represented the threshold probability of the expected benefit of treatment was equal to the expected benefit of avoiding treatment” (Page17, lines 342).

  1. Figure 4. Line 286: Change: “…number who would be declared…” to “…number that would be declared…”

Response: We have changed the word “who” to “that” according to your suggestion (Page17 , lines 346).

  1. Line 286: Change: “…number who would be declared…” to “…number that would be declared…”

Response: We have revised this sentence as responded above.

  1. Put a space between line 291 and line 292.

Response: We have added a space in the corresponding position.

  1. Put a space between line 306 and line 307.

Response: We have added a space in the corresponding position.

  1. Line 323: Change: “…in the diagnosis of and treatment planning…” to “…in the diagnosis of the treatment planning…”.

Response: This paragraph has been moved and merged to introduction.

  1. Line 379: Please remove or close the open parenthesis: “…(95%...”.

Response: We have revised the parenthesis according to your suggestion.

A few additional changes are listed below:

  • Introduction: Add a reference in Line 55 regarding the higher spatial resolution. Maybe reference 29 as cited in the Discussion section.

Response: Thank you again for your kind suggestion. We have added the reference 29 “Michael K O'Connor, et al. EJNMMI Res.2017 Dec 19;7(1):100” regarding the higher spatial resolution in the Introduction section, and it has been reference 29 in the revised manuscript (Page23, line 580-581).

  • Change the y-axis of figures 2 from “Coefficients” to “LASSO coefficients”.

Response: We have changed the y-axis of Figures 2 from “Coefficients” to “LASSO coefficients”. The Figure 2 now shows as below:

  • Figure 4. In subfigure (d) change y-axis from “Number high risk” to “Number of high risk”.

Response: We have changed the y-axis of Figure 4 (d) from “Number high risk” to “Number of high risk”. The Figure 4 now shows as below:

Reviewer 2 Report

In this study, authors aimed to develop a non-invasive integrated machine learning model combining information from dedicated breast PET and clinical characteristics for pathological node status prediction in early-stage breast cancer patients. With this purpose, they enrolled a total of 290 patients, divided into  training and internal validation in  a ratio of 7 to 3, obtaining an NPV of 96.88% in the cN0 subgroup and a PPV of 92.73% in the cN1 subgroup.

Strength:

The topic is undoubtedly of interest and includes an adequate number of patients. The introduction is easy to understand, the methodology used is quite complex, but sufficiently discussed and the results are presented in a clear and exhaustive way.

Weakness:

The so-called dedicated PET (D-PET), including breast PET or “Mammi”-PET and, most of all, axillary lymph node PET, represents an advanced technique, available in a very limited number of tertiary care cancer centres. This represents an important limitation to its applicability in the clinical daily practice.

The modality of radiotracer administration precludes the possibility obtain information about the presence of potential distant metastases and, consequently, to have a more "panoramic" patients’ evaluation.

The absence of an external validation of the impressive results obtained represents an important issue, very difficult to overcome, given the above mentioned scarce availability of scanners provided with this acquisition technique.

In the “Patients” subheading of Methods section (page 2, line 75), authors affirm that this is a prospective study, approved by an Ethics Committee and that the enrolled Patients were randomly divided into two sets (training set and internal validation set). Conversely, in the last sentence of the Discussion section (page 14, line 393), they enumerate the “retrospective design” of the study among the limitations of the study itself. In my opinion this aspect is a bit confusing and need a clarification.

Similarly, authors mention the exclusion of patients with multifocal breast lesions among the limitations of their study. In this regard, it is not clear what they mean with the term of “diffuse tumor location” of some of the patients enrolled.

Author Response

Please see the attachment author responses

In this study, authors aimed to develop a non-invasive integrated machine learning model combining information from dedicated breast PET and clinical characteristics for pathological node status prediction in early-stage breast cancer patients. With this purpose, they enrolled a total of 290 patients, divided into training and internal validation in a ratio of 7 to 3, obtaining an NPV of 96.88% in the cN0 subgroup and a PPV of 92.73% in the cN1 subgroup.

Strength:

The topic is undoubtedly of interest and includes an adequate number of patients. The introduction is easy to understand, the methodology used is quite complex, but sufficiently discussed and the results are presented in a clear and exhaustive way.

Response: Thank you a lot for your positive affirmation of our work.

Weakness:

The so-called dedicated PET (D-PET), including breast PET or “Mammi”-PET and, most of all, axillary lymph node PET, represents an advanced technique, available in a very limited number of tertiary care cancer centres. This represents an important limitation to its applicability in the clinical daily practice.

Response: The dedicated PET is an advanced equipment indeed but it is available because the D-PET scan is open for any surgeons and patients from any other cancer centres and a standard report could be obtained.

The modality of radiotracer administration precludes the possibility obtain information about the presence of potential distant metastases and, consequently, to have a more "panoramic" patients’ evaluation.

Response: The D-PET focused on breast and regional axillary lymph node. The  patient will perform other examination such as CT, bone scan or whole-body PET/CT if necessary to preclude distant metastases.

The absence of an external validation of the impressive results obtained represents an important issue, very difficult to overcome, given the above mentioned scarce availability of scanners provided with this acquisition technique.

Response: Thank you for making this important point. As you pointed out, it is very important to validate the impressive results in an external validation set. As we explained above, the D-PET is available and open for any surgeons and patients from any other cancer centres . In this study, we have validated the results in the individually internal validation set. This study conformed to the transparent reporting of a multivariable prediction model for individual prognosis or diagnosis reporting guidelines.

In the “Patients” subheading of Methods section (page 2, line 75), authors affirm that this is a prospective study, approved by an Ethics Committee and that the enrolled Patients were randomly divided into two sets (training set and internal validation set). Conversely, in the last sentence of the Discussion section (page 14, line 393), they enumerate the “retrospective design” of the study among the limitations of the study itself. In my opinion this aspect is a bit confusing and need a clarification.

Response: In our center, a prospective study (NCT04072653, SOAPET trial) is going on. The SOAPET (Sentinel node biopsy vs. Observation After axillary PET) is a prospective, open-label, phase II clinical trial conducted at Fudan University Shanghai Cancer Center and was divided into two stages. In the first stage, cN0 patients were detected by PE , US and LymphPET, followed by axillary surgery (SLNB or ALND).

Therefore, in this study, patients come from the prospective SOAPET trail (first stage) and a retrospective design and date was analyzed .

Similarly, authors mention the exclusion of patients with multifocal breast lesions among the limitations of their study. In this regard, it is not clear what they mean with the term of “diffuse tumor location” of some of the patients enrolled.

Response: The multifocal lesions mean several separate lesions while diffuse tumor location means a very large lesion. Like the example below.

A: There are 3 separate lesions in one breast gland and that is multifocal lesions.

B: There is a large lesion and extend to the whole gland. That is diffuse tumor. 

A                              B

Reviewer 3 Report

This study is focused on development of a prediction model for the involvement of axial lymph nodes in early stage breast cancer. Input for the prediction model are clinicopathological parameters, and radiomics features derived from dedicated breast PET imaging (mammi-PET). The use of mammi-PET is the main novelty of the paper, and the resulting prediction accuracy is impressively high. The paper is well written, and of interest for the readers of Cancers. It also fits nicely in the special issue of AI in Oncology. I have the following major/minor comments that can be used to improve the paper, and make it acceptable for publication.

Major:

Please remove the term validation from the paper title. Although you have randomly assinged patients to a training and validation set, the validation is not independent, and in some parts of the study the training and validation data has been put together again (see also my later comments).

One of the recurring themes of uncertainty in Radiomics is the interobserver variation in determining the tumor area as input for deriving the imaging features. In this study two independent observers were used to determine the SUVmax on the axilla PET, and to define the tumor on the mammi-PET. In line 227 it is stated that radiomics features were averaged between the two observers, at there was high inter-class correlation. I think this is a waste of efforts. Please also derive a prediction model based on each observer separately, and show how a model trained on one observer validates with the other observer. In the end, if the averaged data between the two outperforms the individual models, you can argue for the need of two independent observers to mitigate interobserver variation.

Abstract: in the conclusion you claim that ML can improve TP and TN, but the reference numbers are not mentioned in the abstract, also not what they are based on.

In the paper conclusion the last line should be toned down, as it is only an expectation that the model will help, but it needs to be proven in an independent dataset.

The introduction is missing some background information on previous work that has been conducted to predict if cN0 is pN0 and if cN1 is a pN1. Especially the study by Bong (28) as mentioned in lines 347-352 is important information. In the discussion the results of study 28 should be compared to the results of the current study. Also, in the discussion the comparison between the current results and previous models is very limited. A short search on pubmed resulted in many potential hits that could be interesting to discuss:

https://pubmed.ncbi.nlm.nih.gov/34991831/

https://pubmed.ncbi.nlm.nih.gov/24475876/

https://pubmed.ncbi.nlm.nih.gov/34739639/

https://pubmed.ncbi.nlm.nih.gov/33421823/

https://pubmed.ncbi.nlm.nih.gov/33122943/

https://pubmed.ncbi.nlm.nih.gov/31226956/

https://pubmed.ncbi.nlm.nih.gov/32256110/

https://pubmed.ncbi.nlm.nih.gov/31372896/

Please do an updated check on existing literature, and discuss the results. The discussion on the performance should also be moved up to the second paragraph of the discussion

The first paragraph of the discussion should summarize the main results of the paper, not repeat the introduction/rationale for the study. In fact, I think parts of the paragraph could be moved to the introduction to introduce the topic better.

Move statistical analysis (156-162) to the end of M&M section

Lines 234 – 236: it is unclear to me if the selection of 4 clinicopathologic markers, 10 Mammi-PET radiomics parameters and 11 combined features were a result of the multivariable regression, or a manual selection. Please make this clearer.

Figure 4: In 4a and 4b the ROCs and AUCs for the different models are shown, with an AUC of the combined model of 0.937 and 0.933 for the training and validation set, respectively. In 4e and 4f a similar ROC curves are shown, but now looking at cN0 and cN1 separately. How come the AUC drops in both graphs to 0.832 and 0.899, while this contains the exact same patients as figure 4a&b? Furthermore, is it actually fair to validate the potential effectiveness of the combined model in the total dataset? This is highly optimistic, as the training set is part of it. I think this should be changed to the validation set only.

Nomogram development in 307: is the nomogram only based on the training set, or is this also based on the total dataset?

Minor comments

Please correct the grammar in lines 69-71.

Line 76: Informed consent was obtained

Line 82: Introduce the term PE

It is not entirely clear to me if the D-PET scan was only performed for patients enrolled in the study, or if it is part of the clinical routine. Would be nice to make clearer to the readers what the study related additional activities were for the patients.

Line 120: single voxel SUV-max, how sensitive is that for noise, and how much noise is present in these scans?

Line 128: cut-off of 0.27 seems to be very low, and arbitrary (as data has not been published). Please validate with other literature on the right cut-off value.

Line 166: introduce the term AIC. I assume that model parameters were only added if they were sufficiently improving the model based on AIC?!

Line 168: introduce the term Pre-scores

Line 173: introduce terms DCA and CIC

Figure 1: why is the mammi-PET not part of the Data collection? And where does the Lymph-PET SUVmax assessment and data use come in?

Table 1: it could also be interesting to statistically compare the training set with the validation set to check if there is an imbalance there.

In lines 332-333 you state that routine ALN status assessment includes US, MMG and MRI. However, in the same paragraph you state many arguments why MMG and MRI are not useful for ALN status. So, are they routine?

Author Response

This study is focused on development of a prediction model for the involvement of axial lymph nodes in early stage breast cancer. Input for the prediction model are clinicopathological parameters, and radiomics features derived from dedicated breast PET imaging (mammi-PET). The use of mammi-PET is the main novelty of the paper, and the resulting prediction accuracy is impressively high. The paper is well written, and of interest for the readers of Cancers. It also fits nicely in the special issue of AI in Oncology. I have the following major/minor comments that can be used to improve the paper, and make it acceptable for publication.

Response: Thank you for your high comment on our work.

Major:

Please remove the term validation from the paper title. Although you have randomly assinged patients to a training and validation set, the validation is not independent, and in some parts of the study the training and validation data has been put together again (see also my later comments).

Response: The “validation” has been removed from the title of the revised manuscript according to your suggestion. The title now reads as follows: “Development of High-resolution Dedicated PET-based Radiomics Machine Learning Model to Predict Axillary Lymph Node Status in Early-stage Breast Cancer” (Page 1, lines 2 to 4).

One of the recurring themes of uncertainty in Radiomics is the interobserver variation in determining the tumor area as input for deriving the imaging features. In this study two independent observers were used to determine the SUVmax on the axilla PET, and to define the tumor on the mammi-PET. In line 227 it is stated that radiomics features were averaged between the two observers, at there was high inter-class correlation. I think this is a waste of efforts. Please also derive a prediction model based on each observer separately, and show how a model trained on one observer validates with the other observer. In the end, if the averaged data between the two outperforms the individual models, you can argue for the need of two independent observers to mitigate interobserver variation.

Response: Thank you for your suggestion. Actually, this effort is unnecessary and has been removed. (Page 11, line 282-284)

Abstract: in the conclusion you claim that ML can improve TP and TN, but the reference numbers are not mentioned in the abstract, also not what they are based on.

Response: For early-stage BC, physical examination and US could delineate clinical subtypes of cN0 and cN1. In cN0 subtype, a higher negative predictive value is needed because the TN lymph node must be identified accurately to omit overtreatment. While in cN1 subtype, a higher positive predictive value is needed because the TP lymph node should be picked out to performed the surgical operation. That is the potential clinical application value existence.

In the paper conclusion the last line should be toned down, as it is only an expectation that the model will help, but it needs to be proven in an independent dataset.

Response: The expression of an expectation that the model will help has been removed. (page 1 line 48)

The introduction is missing some background information on previous work that has been conducted to predict if cN0 is pN0 and if cN1 is a pN1. Especially the study by Bong (28) as mentioned in lines 347-352 is important information. In the discussion the results of study 28 should be compared to the results of the current study. Also, in the discussion the comparison between the current results and previous models is very limited. A short search on pubmed resulted in many potential hits that could be interesting to discuss:

https://pubmed.ncbi.nlm.nih.gov/34991831/

https://pubmed.ncbi.nlm.nih.gov/24475876/

https://pubmed.ncbi.nlm.nih.gov/34739639/

https://pubmed.ncbi.nlm.nih.gov/33421823/

https://pubmed.ncbi.nlm.nih.gov/33122943/

https://pubmed.ncbi.nlm.nih.gov/31226956/

https://pubmed.ncbi.nlm.nih.gov/32256110/

https://pubmed.ncbi.nlm.nih.gov/31372896/

Please do an updated check on existing literature, and discuss the results. The discussion on the performance should also be moved up to the second paragraph of the discussion

Response: Thank you for your professional review and valuable comments. Those literature have been discussed in the discussion and added to the reference of 27, 37-41.

 (Page 20,21, 23 and 24. Lines 409-411, 446-452, 576-577, 598-607).

The first paragraph of the discussion should summarize the main results of the paper, not repeat the introduction/rationale for the study. In fact, I think parts of the paragraph could be moved to the introduction to introduce the topic better.

Response: The first paragraph of discussion has been moved to introduce. (Page 2, Lines 62-66).

Move statistical analysis (156-162) to the end of M&M section

Response: Thank you again for your kind suggestion. We have moved the “statistical analysis” to the end of M&M section (Page 8, lines 230-236).

Lines 234 – 236: it is unclear to me if the selection of 4 clinicopathologic markers, 10 Mammi-PET radiomics parameters and 11 combined features were a result of the multivariable regression, or a manual selection. Please make this clearer.

Response: The selection of 4 clinicopathologic markers, 10 Mammi-PET radiomics parameters and 11 combined features were a result of the multivariable regression. We have revised the sentence to make it clearer as follow: “3 independent prediction models were developed using the most valuable 4 clinicopathologic markers, 10 Mammi-PET radiomics parameters, and 11 combined features selected by the multivariable regression with the AIC” (Page 11, lines 289-292).

Figure 4: In 4a and 4b the ROCs and AUCs for the different models are shown, with an AUC of the combined model of 0.937 and 0.933 for the training and validation set, respectively. In 4e and 4f a similar ROC curves are shown, but now looking at cN0 and cN1 separately. How come the AUC drops in both graphs to 0.832 and 0.899, while this contains the exact same patients as figure 4a&b? Furthermore, is it actually fair to validate the potential effectiveness of the combined model in the total dataset? This is highly optimistic, as the training set is part of it. I think this should be changed to the validation set only.

Response: The Figure 4a and 4b showed the ROCs and AUCs for the different models in discriminating pN0 from pN1 patients in the training and validation set, respectively. The results indicated the Combined Model held the greatest prediction performance (AUCs of 0.937 and 0.933, respectively). The Figure 4e and 4f showed the effectiveness of the Combined Model in identifying pN0 from cN0 patients with an AUC of 0.83, while identifying pN1 from cN1 patients with an AUC of 0.90. Different AUCs due to the different applications of the Combined Model. Otherwise, as you pointed out, it is very important to validate the potential effectiveness of the Combined Model in the independent validation set. We will continue to collect a sufficient number for subsequent validation.

Nomogram development in 307: is the nomogram only based on the training set, or is this also based on the total dataset?

Response: The nomogram was developed based on the training set, as the previous studies (Huang YQ et al. J Clin Oncol. 2016;34(18):2157-2164; Ren CY et al. Eur J Nucl Med Mol Imaging. 2021;48(5):1538-1549). It also has been definite in the Methods section (Page 8, lines 224-229).

Minor comments

Please correct the grammar in lines 69-71.

Response: We have rephrased the sentence in the revised manuscript as follows: “This study aimed to develop a scalable, non-invasive and robust machine learning model for the prediction of pNx, which was termed radiomics lymph node status (rNx) using D-PET radiomics integrated the clinical characteristics in early-stage BC. ” (Page 2, lines 85-87).

Line 76: Informed consent was obtained

Response: done. (page 2 line 93)

Line 82: Introduce the term PE

Response: done (page 3 line 100)          

It is not entirely clear to me if the D-PET scan was only performed for patients enrolled in the study, or if it is part of the clinical routine. Would be nice to make clearer to the readers what the study related additional activities were for the patients.

Response: In our center, a prospective study (NCT04072653, SOAPET trial) is going on. The SOAPET (Sentinel node biopsy vs. Observation After axillary PET) is a prospective, open-label, phase II clinical trial conducted at Fudan University Shanghai Cancer Center and was divided into two stages. In the first stage, cN0 patients were detected by PE , US and LymphPET, followed by axillary surgery (SLNB or AD).

Therefore, in this study, D-PET scan was performed for patients enrolled in SOAPET trail and the retrospective date also come from the first stage of SOAPET trail.

Line 120: single voxel SUV-max, how sensitive is that for noise, and how much noise is present in these scans?

Response: The dedicated LymphPET possesses is the high -resolution because of the following mechanism. Unlike WB-Whole body PET/CT, which has a large gantry where the PET ring is mounted, LymphPET is designed to use bi-planar detectors to observe specific regions of interest (ROIs). For example, when it is used to detect the metastatic status of axillary lymph nodes, the patient’s axilla is positioned in the middle of the bi-planar detectors, and the detectors close to the patient lead to a higher sensitivity.  Each square- shaped detector plane is designed 20x20 cm in size, and it is composed of 16 units of “double-sided front-end readout modules (DRM) ” which integrates the LYSO crystal arrays, two 2 SiPM arrays, and thecomposed of 16 units of "double-sided front-end readout modules (DRM)" that integrate the LYSO crystal arrays and 2 SiPM arrays frontend electronics in a compact detector module. In order to deliver better sensitivity, spatial resolution, and timing resolution, an Silicon photomultiplier (SiPM) is used instead of a traditional photomultiplier tube (PMT) . By measuring light signals to compare the ratio between light outputs detected at one side and both sides, the double-sided module design has an advantage to in providing continuous depth-of- interaction (DOI) information with better DOI resolution. DOI information will not only improve time resolution, but will also improve time resolution and enhance spatial resolution, even at the center of the field of vision (FOV) .

For the diagnostic performance of LymphPET, please refer to our article “Diagnostic performance of a novel high-resolution dedicated axillary PET system in the assessment of regional nodal spread of disease in early breast cancer” https://dx.doi.org/10.21037/qims-21-388

Line 128: cut-off of 0.27 seems to be very low, and arbitrary (as data has not been published). Please validate with other literature on the right cut-off value.

Response:  The cut-off of 0.27 also come from our article “Diagnostic performance of a novel high-resolution dedicated axillary PET system in the assessment of regional nodal spread of disease in early breast cancer” https://dx.doi.org/10.21037/qims-21-388

Line 166: introduce the term AIC. I assume that model parameters were only added if they were sufficiently improving the model based on AIC?!

Response: The Akaike Information Criterion (AIC) is a well-known tool for variable selection in multivariable model as well as a tool to help identify the optimal representation of explanatory variables. The AIC can be employed to consider models based on all possible subsets of explanatory variables, unlike certain other model selection techniques that only consider a few potential models among the entire candidate collection (e.g., backward elimination, forward selection) (VanBuren J et al. Journal of Public Health Dentistry. 2017;77(4):360-71).

In this study, 7 of the 12 clinicopathologic markers, 34 of the 851 radiomics features, and 19 of the 864 combined features were initially selected by the LASSO regression. Actually, inclusion of more covariates does not necessarily lead to higher accuracy, but rather to overfitting and should be avoided. Finally, the most valuable 4 clinicopathologic markers, 10 Mammi-PET radiomics parameters, and 11 combined features were selected to develop the prediction models by the AIC.

Line 168: introduce the term Pre-scores

Response: The Pre-score was calculated using the linear fusion of the selected non-zero features and their coefficients. We have supplemented the details of Pre-score in the Methods section of the revised manuscript. The text now reads as follows: “the prediction score (Pre-score) was calculated for each patient using the linear fusion of the selected non-zero features and their coefficients” (Pages 7, lines 211-213).

Line 173: introduce terms DCA and CIC

Response: The DCA is a graphical summary proposed for assessing the potential clinical impact of risk prediction biomarkers or risk models for recommending treatment or intervention. The DCA is grounded in a decision-theoretical framework that accounts for both the benefits of intervention and the costs of intervention to a patient who cannot benefit. It shows the performance of a risk model in a population in which every patient has the same expected benefit and cost of intervention. The CIC is another type of plot produced by the same R package (“DecisionCurve”) with DCA. For a single-risk model, the CIC shows the estimated number who would be declared high risk for each risk threshold and visually shows the proportion of those who are cases (true positives) (Kerr KF et al. J Clin Oncol. 2016;34(21):2534-40).

In this study, the DCA showed that the Combined Model was the most reliable and valuable tool to predict the ALN status when the ALN metastasis (ALNM) threshold probability was greater than 10%. The CIC of the Combined Model presented the risk stratification in predicting 1000 people, including the estimated number of people who would be declared a high risk for ALN metastasis and true positive cases under each threshold probability.

Figure 1: why is the mammi-PET not part of the Data collection? And where does the Lymph-PET SUVmax assessment and data use come in?

Response: Mammi-PET only provides the images of the primary breast tumor for the tumor segmentation and quantitative radiomics feature extraction, as shown in the part of the PET image preprocessing in the Figure 1. We have also added this detail to make it clearer in the Methods section of the revised manuscript as follow: “Tumor was visualized and segmented on the Mammi-PET images” (Page 7, lines 173-174).

Lymph-PET images show whether there are lymph nodes with 18F-FDG uptake in the bilateral axillary regions. In this study, the SUVmax cutoff value of lymph node on Lymph-PET was set at 0.27. SUVmax of ≥0.27 was considered positive, while that of <0.27 was considered negative. The assessment of lymph node on the Lymph-PET images were collected and used in the subsequent analysis.

Table 1: it could also be interesting to statistically compare the training set with the validation set to check if there is an imbalance there.

Response: 290 patients were randomly divided into the training (n=203, 70.00%) and validation (n=87, 30.00%) sets according to the ratio of 7 to 3. There were 94 patients (46%) with pN0 and 109 patients (54%) with pN1 in the training set, while 49 patients (56%) with pN0 and 38 patients (44%) with pN1 in the validation set. The positive and negative samples in the two sets were balanced.

In lines 332-333 you state that routine ALN status assessment includes US, MMG and MRI. However, in the same paragraph you state many arguments why MMG and MRI are not useful for ALN status. So, are they routine?

Response: Thank you for your kind suggestion. The more proper expression is follow: “Routine primary lesion and ALN status assessment includes US, mammography (MMG), and magnetic resonance imaging (MRI). Although US remains one of the key tools for ALN assessing, it has the limitation of being subjective”. (Page 20, line 394-396)

Round 2

Reviewer 1 Report

I propose the acceptance of the manuscript

Author Response

No notes

Reviewer 3 Report

The paper nicely improved compared to the last version, and all of my comments have been addressed properly. I do have some final minor remarks that need to be addressed:

Please remove the word 'remarkable' from describing the performance in the simple summary and the conclusions. I think it is up to the reader to call it remarkable or not. 

Please shorten the conclusion of the paper, as there is now a lot of technical detail present, and repeat of information (AUC is reported twice). Report percentages with only one digit after the ., and remove specific details as the optimal threshold of 0.59, as this is very specific for the developed nomogram.

Author Response

Manuscript ID: Cancers - 1528297

Title: Development and Validation of High-resolution Dedicated Positron Emission Tomography-based Radiomics Machine Learning Model to Predict Axillary Lymph Node Status in Early-stage Breast Cancer

Responses to the comments of Reviewer 3 (round 2)

Reviewer 3:

The paper nicely improved compared to the last version, and all of my comments have been addressed properly. I do have some final minor remarks that need to be addressed:

Response: Thank you for your high comment on our work.

Minor:

Please remove the word 'remarkable' from describing the performance in the simple summary and the conclusions. I think it is up to the reader to call it remarkable or not. 

Response: The “remarkable” has been removed from describing the performance in the simple summary and conclusions. (Page 1).

Please shorten the conclusion of the paper, as there is now a lot of technical detail present, and repeat of information (AUC is reported twice).

Response: The conclusion has been shorten and some repeat sections have been removed. (Page 13).

Report percentages with only one digit after the ., and remove specific details as the optimal threshold of 0.59, as this is very specific for the developed nomogram.

Response: All percentages have been checked again. The description of “rNx and 0.59” has been deleted and corrected.
